# Wavelet-Transform-Based Sparse Code Multiple Access for Power Line Communication

**Muhammad Sajid Sarwar** [1,†] , **Sobia Baig** [1] , **Hafiz M. Asif** [2,*] **and Kaamran Raahemifar** [3,4,5] **and Samir Al-Busaidi** [2]

1 Department of Electrical and Computer Engineering, COMSATS University Islamabad, Lahore Campus, Punjab 54000, Pakistan
2 Department of Electrical and Computer Engineering, Sultan Qaboos University, Muscat 123, Oman
3 Data Science and Artificial Intelligence Program, College of Information Sciences and Technology (IST), Pennsylvania State University, State College, PA 16801, USA
4 Faculty of Science, School of Optometry and Vision Science, University of Waterloo, 200 University Ave. W, Waterloo, ON N2L 3G1, Canada
5 Department of Chemical Engineering, Faculty of Engineering, University of Waterloo, 200 University Ave. W, Waterloo, ON N2L 3G1, Canada
* Correspondence: h.asif@squ.edu.om
† Current address: Wireless and Emerging Network System (WENS) Lab, Kumoh National Institute of Technology, Gumi 39177, Korea.

**Abstract:** This paper presents Discrete Wavelet Transformed Sparse Code Multiple Access (DWT-SCMA) in Power Line Communication (PLC) systems. In the present internet of things era, PLC provides an established infrastructure for low-cost and reliable indoor connectivity. PLC systems can benefit from the Sparse Code Multiple Access (SCMA) technique, which allows multiple users to access a frequency slot simultaneously to maximize spectrum efficiency. However, interuser interference arises in SCMA when numerous users map their data to the same frequency resource; this, in turn, is likely to be enhanced by the noisy PLC channel. This article adopts the intriguing aspects of DWT to address the interference difficulties. A mathematical model of the proposed technique is also presented and compared with Fast Fourier Transformed SCMA (FFT-SCMA). In the PLC environment, DWT-SCMA is found to outperform FFT-SCMA.

**Keywords:** Fast Fourier Transform; interuser interference; nonorthogonal multiple access; orthogonal frequency division multiplexing; Power Line Communication; pulse-shaping schemes; Sparse Code Multiple Access; discrete wavelet transform

## 1. Introduction

Future communication systems are expected to bring a paradigm shift, especially with reference to system speed, user capacity, latency, and energy efficiency. Power Line Communication (PLC) is an efficient, robust, and cost-effective data communication technology that provides an extensive coverage area by utilizing existing electrical power transmission infrastructure. PLC is popular for smart grid and in-home networking applications. It has become significant in many indoor scenarios, such as basements and buildings with metal or concrete walls, where wireless signals suffer from high losses. As the basic purpose of power lines is not to carry communication signals, PLC systems have major challenges, which include non-Gaussian noise, frequency selectivity, and low signal power due to electromagnetic compatibility issues. Based on data speeds, PLC is categorized into three different technologies, namely, ultra-narrowband PLC, narrowband PLC, and Broadband PLC (BPL) [1]. We refer to the case of BPL for communication system evaluation and design in this research work. BPL aims for higher data rates, and several Multiple Access (MA) schemes for this technology are discussed in the literature, most of which are Orthogonal Multiple Access (OMA) techniques [1,2]. Orthogonal Frequency Division Multiple Access

(OFDMA) deals with non-Gaussian noise and the frequency-selective channel of PLC. The spectrum efficiency of OFDM subchannels can be further increased with the application of Nonorthogonal Multiple Access (NOMA) technique, which is validated from recent research, especially for multiuser cases [2].

The type of Multiple Access (MA) scheme, applied to a particular generation of communication systems, has special significance in differentiating the performance of various communication systems [3]. The conventional MA techniques that encompass the OMA schemes employ orthogonal resources either in the time, frequency, or code domain to avoid interuser interference. Even though OMA approaches have relatively lower complexity, they have limited performance in terms of user connectivity and throughput. Hence, the OMA techniques are unable to meet the requirements of the future communication era of the internet of things. To address the challenges, some powerful solutions, for instance, massive multiple-input multiple-output, millimeter wave communications, ultradense networks, and NOMA have been proposed in the literature [4,5].

NOMA is predicted to become popular in beyond fifth-generation cellular communications because of efficient spectrum utilization and cost-effectiveness [6]. It is useful for the PLC environment as well to support different devices for communication at different data speeds [2]. In NOMA techniques, simultaneous users can transmit data over the same frequency but at different power levels or by using multiple codes [7]. NOMA is a spectral efficient system without compromising the data rates. There are various types of NOMA, including Power-Domain (PD) NOMA and Code-Domain (CD) NOMA. PD-NOMA and CD-NOMA are differentiated by the user multiplexing method. The former employs power-domain multiplexing while the latter utilizes code-domain multiplexing [8]. PD-NOMA is suggested as a useful technique for wireless channel [9]. To cope with modern demands of massive connectivity and spectrum incompetency of OMA, NOMA can also be applied in PLC as well [2–10]. Research works related to PLC-NOMA can be found in [10–12]. Signals from several users are superimposed in this method at the same frequency and time slot but with different power levels based on their relative distance from the base station. This superimposed signal is then transmitted, and at the receiver, a Successive Interference Cancellation (SIC) helps to separate the signal for the intended user. The performance of SIC deteriorates in terms of user detection and throughput as the number of users increases. Sparse Code Multiple Access (SCMA) can provide a solution for high connectivity as compared with PD-NOMA with desirable data rates [8]. It employs the nonorthogonality of the codes. It is one of the CD-NOMA techniques that use three-dimensional codebooks, designed from the rotation of the constellation [13]. Each user is assigned a separate codebook for data encoding [14]. Data from multiple users are encoded using individual sparse codebooks and then superimposed for transmission. A complex receiver, such as a Message Passing Algorithm (MPA) based on maximum likelihood detection, is utilized to extract the desired signal [15]. The SCMA provides massive connectivity with higher data rates, but it also increases Interuser Interference (IUI) within the SCMA system [16–19].

Multicarrier (MC) transmission is a prominent multiplexing and modulation technology in digital communications. Single-carrier modulated transmission may result in the loss of the entire data packet if the data are distorted owing to noise in the channel. MC transmission, on the other hand, provides flexibility and reliability in data transmission. The PLC channel is considered inherently a noisy channel and is also prone to electromagnetic interference. Therefore, MC transmission—i.e., OFDM—is beneficial in terms of robustness and reliability of communication [1]. Different standards for PLC are available in literature, such as IEEE 1901, which utilizes Fast Fourier Transformed (FFT) Orthogonal Frequency Division Multiplexing (FFT-OFDM) as well as Discrete Wavelet Transform (DWT) OFDM [1]. As power cables were originally designed to transmit power instead of data, the communication medium is vulnerable to channel impairments such as frequency selective fading [20]. One of the most significant interferences for PLC systems is harmonic oscillation induced by switching devices [21]. SINR decreases as a result of the certain noise level of high-frequency harmonics. Although FFT is one of the simplest and most effective



techniques for harmonic detection, it has drawbacks such as spectrum leakage, challenges with nonstationary signal analysis, and only provides frequency domain visualization. Unlike the FFT, the DWT's time–frequency localization properties can adapt to the changes. DWT is frequently employed in the harmonic analysis of power systems because it can maintain all of the temporal and frequency information of the signal [22]. The use of DWT results in a reduction in noise. Furthermore, by splitting the spectrum to enable multiresolution analysis, DWT minimizes computation complexity compared with FFT, which performs computation at every frequency point [23,24]. DWT has a pulse shape with low power in side-lobes, which helps to reduce Intersymbol Interference (ISI) and ICI without appending a Cyclic Prefix (CP), making it spectral-efficient [25,26]. DWT-OFDM for PLC was investigated in [22,27] and shown to outperform FFT-OFDM in terms of BER. FFT-OFDM and DWT-OFDM can also be integrated with SCMA to enhance their performance. In this document, FFT-OFDM-based SCMA is termed as FFT-SCMA, whereas SCMA employing DWT-OFDM will be referred to as DWT-SCMA for the PLC channel. The scenario without pulse shaping in wireless and PLC environments is represented in this article by SCMA and PLC-SCMA, respectively.

The spectrum efficiency of NOMA is affected by IUI. The IUI can be reduced with pulse-shaping techniques such as FFT and DWT. FFT-OFDM has been combined with PD-NOMA (FFT-NOMA) [28] and SCMA [29], resulting in higher data rates and improved BER performance. FFT-SCMA has several intrinsic issues such as a high Peak-to-Average-Power Ratio (PAPR), which is harmful to power amplifiers, spectral inefficiency due to the use of CP, and limited data visualizations of FFT to handle the interference appropriately [29]. To obtain better data decomposition and reconstruction as compared with FFT, there are other methods such as DWT and Wavelet Packet Transform (WPT) [30,31]. The transceiver, employing the DWT-OFDM scheme for resource allocation in PD-NOMA, was proposed in [32,33]. DWT-NOMA provides better performance as compared with conventional FFT-NOMA in terms of interference mitigation, multiuser capacity, spectrum confinement, and efficiency [28–33]. In [30], cognitive radio capabilities were explored for the SCMA system. In order to reduce mutual interference between other operating networks, such as WiGig and cognitive users for a 5G network, this system incorporated cognitive traits including spectrum sensing, interference estimates, and subcarrier adaption. Instead of Fourier-transform-based spectrum sensing, Wavelet Packet Transform (WPT) spectrum sensing was employed to conduct an improved estimation of frequency holes. This article presents DWT-SCMA for PLC systems, which is the first study of its kind to the best of our knowledge.

### 1.1. Contribution

To the best of our knowledge, no research on interference reduction for PLC-SCMA has been conducted in the literature. Following are the major contributions of this research work:

- In this paper, the authors suggest DWT-SCMA for the PLC system. Instead of WPT, DWT is adopted due to its sparse structure.
- FFT-SCMA uses MC transmission to reduce multipath fading effects and IUI, although it still has Intercarrier Interference (ICI). The pulse-shaping with DWT instead of FFT can lower ICI, resulting in a higher Signal-to-Interference-Plus-Noise Ratio (SINR).
- In this work, a mathematical expression for SINR is derived for FFT-SCMA and DWT-SCMA. The impact of both techniques on noise and interference is demonstrated through mathematical equations.
- The authors have computed numerical results of FFT-SCMA and DWT-SCMA by taking the PLC channel into consideration. SINR, throughput, PAPR, and Symbol-Error-Rate (SER) are used to evaluate performance in the PLC channel, revealing that DWT-SCMA outperforms FFT-SCMA.

*1.2. Organization*

Section 2 describes the system model. It explains the conventional PLC channel, the mathematical model for PLC-SCMA, the IUI problem, FFT-SCMA, and DWT-SCMA for PLC in Sections 2.1, 2.2, 2.4, 2.5, and 2.6, respectively. In Sections 2.7–2.9, the mathematical expressions of capacity, SER, and PAPR are described, respectively. Section 3 discusses the results of SINR, capacity, PAPR, and SER for FFT-SCMA and DWT-SCMA applications in a PLC environment with and without pulse-shaping mechanisms. The article is concluded in Section 4.

## 2. System Model Description

*2.1. Channel Response of PLC*

Power cables employed for communication are originally connected to various load impedances, which can give rise to impedance mismatch at different branches, resulting in reflected versions of the transmitted signal. Impulsive noise, synchronous and asynchronous noise, narrowband noise, and attenuation are significant characteristics of the power line channel. Mathematical and simulation models presented in various research works for PLC demonstrate that a power line channel has narrowband fading, as depicted in Figure 1 [20,34]. The PLC channel frequency response appears selectively faded with narrowband notches over the entire frequency range. The complex transfer function of the PLC channel, which specifies the universally applicable and practical form, is expressed as [35],

$$H(f) = \sum_{d=1}^{D} v_d e^{-(a_0 + a_1 f^z) p_d} \cdot e^{-\frac{j2\pi f p_d}{v}} \tag{1}$$

where the first and second exponentials signify the attenuation and echo scenarios, respectively. The model is defined by the superposition of signals from $D$ different paths, each with a $v_d$ weighting factor, $p_d$ length, and $v$ propagation speed. Frequency-dependent attenuation is modeled using parameters derived from the magnitude of the frequency response, such as attenuation offset $a_0$, attenuation increase $a_1$, and exponent of attenuation $z$.

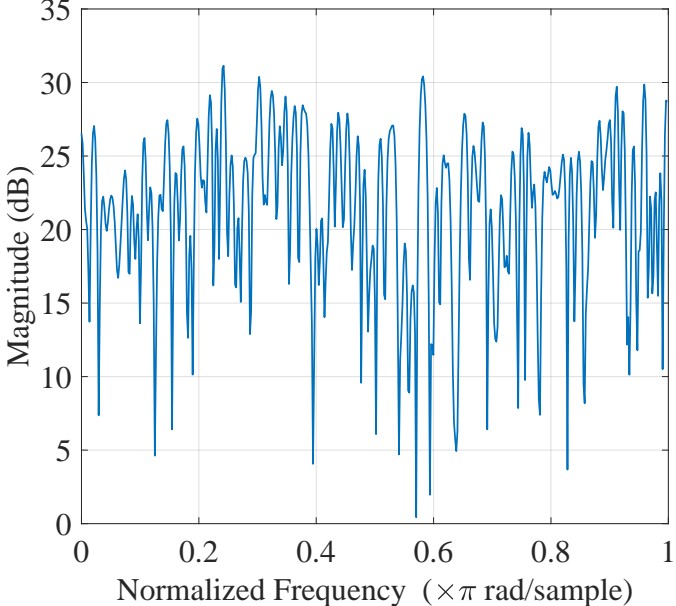

**Figure 1.** PLC channel frequency response.

In PLC-SCMA, FFT-OFDM signals employ Sine and Cosine basis functions to split a wide channel into multiple subchannels, for which the PLC channel's narrowband fading seems to be flat fading. DWT-OFDM signaling for PLC-SCMA is comparatively better than

FFT-OFDM signal because it utilizes basis functions such as Shannon or Symlet, which are less prone to sidelobe attenuation and help to minimize noise and interference.

### 2.2. SCMA System Model for PLC

SCMA utilizes sparse resource (frequency) allocation and nonorthogonal three- dimensional codebooks to increase spectrum efficiency. The encoded data is mapped on available frequency resources in a sparse manner to support overload conditions. Figure 2a presents a factor graph that elaborates the user mapping on resources. Assume that $K$ represents the number of users and $N$ denotes the number of subchannels or resources. The overloading factor is given as $K/N$, with $K > N$. In the factor graph, each user node is referred to as a "Variable Node (VN)", whereas a resource is known as a "Function Node (FN)". The following matrix $\mathbf{F}$ represents an example of a sparse mapping between FN and VN for $K = 6$ users and $N = 4$ resources.

$$\mathbf{F} = \begin{bmatrix} 0 & 1 & 1 & 0 & 1 & 0 \\ 1 & 0 & 1 & 0 & 0 & 1 \\ 0 & 1 & 0 & 1 & 0 & 1 \\ 1 & 0 & 0 & 1 & 1 & 0 \end{bmatrix} \tag{2}$$

Each row of the matrix $\mathbf{F}$ contains three nonzero elements representing multiple users; the data on a resource correspond to the row number. In this matrix, $d_v = 2$ and $d_f = 3$, which denote the VN degree and FN degree, respectively. $d_v$ represents the number of subchannels employed by one user. On the other hand, the function node degree $d_f$ indicates the number of users that a resource can accommodate. The ones of the $\mathbf{F}$ matrix will be replaced by channel coefficients in the case of Independent Identically Distributed (IID) Additive White Gaussian Noise (AWGN). Now, the channel matrix $\mathbf{H}_s$ that corresponds to the factor graph can be written as follows:

$$\mathbf{H}_s = \begin{bmatrix} 0 & h_{1,2} & h_{1,3} & 0 & h_{1,5} & 0 \\ h_{2,1} & 0 & h_{2,3} & 0 & 0 & h_{2,6} \\ 0 & h_{3,2} & 0 & h_{3,4} & 0 & h_{3,6} \\ h_{4,1} & 0 & 0 & h_{4,4} & h_{4,5} & 0 \end{bmatrix} \tag{3}$$

The channel matrix $\mathbf{H}_s$ for six users and four frequency resources is made up of twelve distinct links that represent an IID AWGN scenario. The channel coefficients $h_{n,k} \in \mathbf{H}_s$ for PLC system are characterized by signal attenuation, delay, and phase shift, as discussed in Section 2.1. Each channel coefficient in the matrix $\mathbf{H}_s$ corresponds to a link between $n$th resource and $k$th user such as $h_{n,k}$, which is expressed as follows:

$$\mathbf{H}'_s = \begin{bmatrix} 0 & 0 & h_{1,2} & 0 & h_{1,3} & 0 & 0 & 0 & h_{1,5} & 0 & 0 & 0 \\ h_{2,1} & 0 & 0 & 0 & 0 & h_{2,3} & 0 & 0 & 0 & 0 & h_{2,6} & 0 \\ 0 & 0 & 0 & h_{3,2} & 0 & 0 & h_{3,4} & 0 & 0 & 0 & 0 & h_{3,6} \\ 0 & h_{4,1} & 0 & 0 & 0 & 0 & 0 & h_{4,4} & 0 & h_{4,5} & 0 & 0 \end{bmatrix} \tag{4}$$

The rotated constellation diagram is used to generate nonorthogonal complex codes, which are then multiplied by user data. The following equation can be used to represent the SCMA codebook design [13],

$$\mathbf{G} = \arg \max_g m(S(\mathbf{H}_s, N, K, d_v, \mathbf{C}, M)) \tag{5}$$

where $M$ is the order of modulation for each user, $S(.)$ represents the matrix function related to the variables in (5), and $m(.)$ gives the performance measure in the codebook design. Here, $\mathbf{C}$ denotes a code-word matrix that consists of complex numbers obtained from the rotation of the constellation diagram (i.e., QPSK or QAM), as shown in Figure 2b.

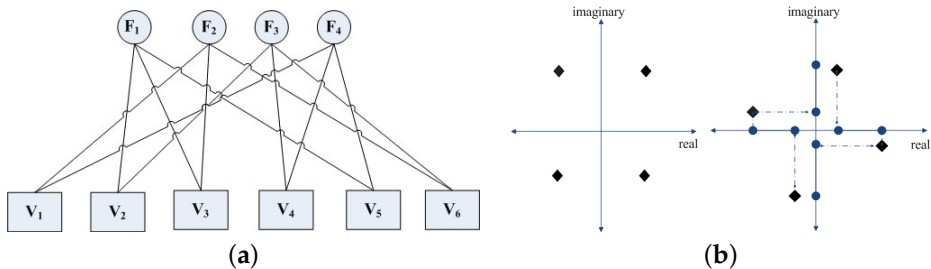

**Figure 2.** (**a**) Factor graph representation. (**b**) Constellation Rotation for Codebook design in SCMA.

**C** can be written based on the following equation:

$$c_{ij} = A \cdot \sin(2\pi + \theta_{ij})i + A \cdot \cos(2\pi + \theta_{ij}) \tag{6}$$

where $c_{ij}$ is an element of the matrix **C**, whereas $A$ and $\theta_{ij}$ represent the amplitude and the rotation angle for a code-word, respectively. For the fourth-order modulation, the amplitudes of 00, 01, 10, and 11 are set to $-0.2243$ V, $-0.7851$ V, $0.2243$ V, and $0.7851$ V, respectively. Here, **C** takes the following form (referring to (6)):

$$\mathbf{C} = \begin{bmatrix} -0.2243 & -0.0055 - 0.2243i \\ -0.7851 & 0.0193 - 0.7848i \\ -0.1815 - 0.1318i & -0.1392 - 0.1759i \\ 0.6315 - 0.4615i & 0.4873 - 0.6156i \end{bmatrix} \tag{7}$$

On the basis of **C**, six different complex sparse codebooks can be designed as given in [36]. A graphical representation of a PLC-SCMA encoder and user codebooks is depicted in Figure 3. Here, six codebooks are illustrated with two nonzero entries in each column. The length of each column is four, which is equal to the number of available resources. Each codebook is multiplied by the user data based on the number of bits required to send for that user. The decoder receives this encoded data, after it has been superimposed and transmitted through the channel, as presented in Figure 3.

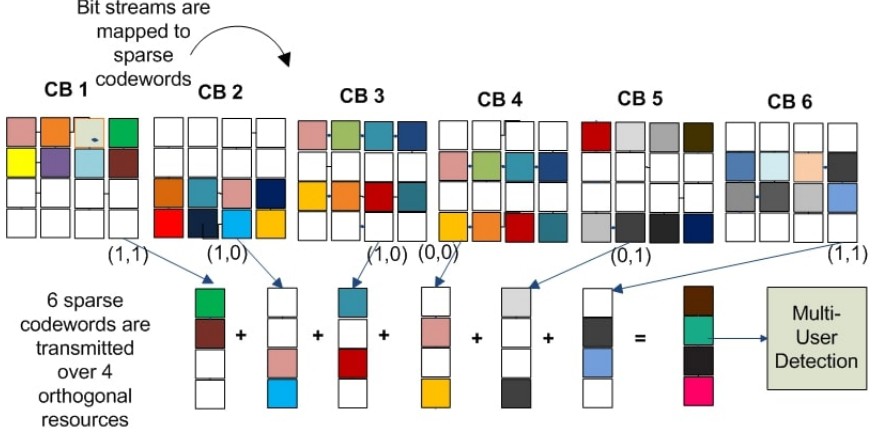

**Figure 3.** Graphical representation of an SCMA encoder.

The PLC-SCMA encoded signal is represented by $\mathbf{x} = \mathbf{Gb}$, where **G** and **b** denote the codebook and the information bits, respectively. The received signal **y** is written below,

$$\mathbf{y} = \mathbf{H}_s' \mathbf{x} + \mathbf{w} \tag{8}$$

where $\mathbf{H}_s'$ represents the channel matrix in (4) and **w** is IID AWGN with zero mean and variance $\sigma^2$.

### 2.3. Decoding for SCMA

The traditional MPA decoder is used in this study, which is well-known in the literature and can be found in [37–41]. Since no novel decoder is suggested in this investigation, the MPA is just given a brief. The flow diagram of the MPA decoder is illustrated in Figure 4. The MPA assists in extracting data from each user after the factor graph is known and assumes that the receiver possesses Channel-State-Information (CSI). The FN (resources) and VN (users) iteratively pass messages in the form of log probabilities along the edges of the factor graph based on the received signal **y**. The messages passing from VN to UN are initialized first as $V_{tv}[\mathcal{G}] = 0$, where $\mathcal{G} \in \mathbf{G}$. The messages delivered from FN to VN are written as [38]

$$U_{vf}[\mathcal{G}] = \max_{v \in N} \left( L_v[\beta] + \sum_{t=1, t \neq f}^{d_f} V_{tv}[\mathcal{G}] \right) \atop \mathcal{G} \in \mathbf{G}, \quad t = 1, \dots, d_f. \tag{9}$$

where $L_v[\mathcal{G}]$ is defined below [38]

$$L_v[\mathcal{G}] = \frac{1}{\sigma^2} \left\| y^n - \sum h_{n,k} x_{n,k} \right\| \tag{10}$$

The channel and encoded symbol between the $k$th user and the $n$th resource are represented by $h_{n,k}$ and $x_{n,k}$, respectively. $\sigma^2$ denotes the noise variance and $y^n$ is the received signal at $n$th resource. Furthermore, VNs are updated as $V_{fv}[\beta] = \sum_{t=1, t \neq f}^{d_v} U_{tf}[\mathcal{G}]$ and the decision is made as $L_f[\hat{\mathcal{G}}] = \max_{f \in K} \left( \sum_{t=1}^{d_f} U_{tf}[\mathcal{G}] \right)$.

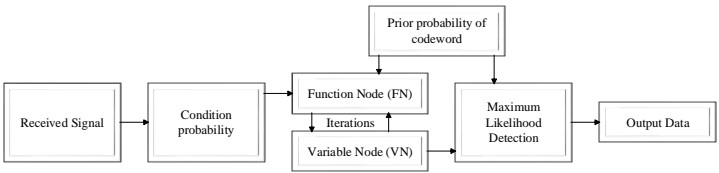

**Figure 4.** Block diagram of an MPA decoder.

### 2.4. Interuser Interference in PLC-SCMA

The IUI of the PLC-SCMA system model is presented in this section. A scenario with $K = 6$, $N = 4$, $d_v = 2$, and $d_f = 3$ is considered as an example. $x_{n,k} \in \mathbf{x}$ signifies the symbol for $k$th user over $n$th resource, which can be specified with the aid of (2); the encoded vector **x** is written as follows:

$$\mathbf{x} = \begin{bmatrix} x_{2,1} & x_{4,1} & x_{1,2} & x_{3,2} & x_{1,3} & x_{2,3} & x_{3,4} & x_{4,4} & x_{1,5} & x_{4,5} & x_{2,6} & x_{3,6} \end{bmatrix}^T \tag{11}$$

For the abovementioned scenario, **y** in (8) can be expanded as follows:

$$\mathbf{y} = \begin{bmatrix} y_1 \\ y_2 \\ y_3 \\ y_4 \end{bmatrix} = \begin{bmatrix} h_{1,2}x_{1,2} + h_{1,3}x_{1,3} + h_{1,5}x_{1,5} \\ h_{2,1}x_{2,1} + h_{2,3}x_{2,3} + h_{2,6}x_{2,6} \\ h_{3,2}x_{3,2} + h_{3,4}x_{3,4} + h_{3,6}x_{3,6} \\ h_{4,1}x_{4,1} + h_{4,4}x_{4,4} + h_{4,5}x_{4,5} \end{bmatrix} + \begin{bmatrix} w_{1,2} \\ w_{2,1} \\ w_{3,2} \\ w_{4,1} \end{bmatrix} \tag{12}$$

This case is termed an overloaded scenario, where $K > N$ and the interference observed by the single user are computed here. In the given case, each frequency resource allows three users to map their data at a time; so, every resource has one desired user with two interferers. For instance, user-to-resource mapping is depicted in Figure 2a with the factor graph representation, where data from user node $V_1$ are mapped on $F_2$ along with the data of $V_3$ and $V_6$. Similarly, $F_4$ also receives data of $V_1$ in addition to $V_4$ and $V_5$. Maximal-Ratio-Combining (MRC) is applied on the received signal **y** in (12) to combine the

signal of the *k*th user (for example, User 1, in this case) for the analysis of the effect of IUI over a single user, as expressed here,

$$
\begin{aligned}
y' = &|h_{2,1}|^2 x_{2,1} + |h_{4,1}|^2 x_{4,1} + h_{2,1}{}^* h_{2,3} x_{2,3} + h_{2,1}{}^* h_{2,6} x_{2,6} \\
&+ h_{4,1}{}^* h_{4,4} x_{4,4} + h_{4,1}{}^* h_{4,5} x_{4,5} + w_{2,1} h_{2,1}{}^* + w_{4,1} h_{4,1}{}^*
\end{aligned}
\tag{13}
$$

where $y' = y_2 h_{2,1}^* + y_4 h_{4,1}^*$ in (13) represents the received signal strength for a user, from which received SINR is defined as follows:

$$
SINR_l = \frac{S}{I+N} = \frac{E_s * h'^2}{E_s\, H + h'^2 \sigma^2}
\tag{14}
$$

where $h'$ is related to the desired signal, $H$ shows IUI, and $\sigma^2$ indicates noise variance. These are given in Equations (15), (16), and (17), respectively.

$$
h' = |h_{2,1}|^2 + |h_{4,1}|^2
\tag{15}
$$

$$
H = \left( \left|h_{2,1}^* h_{2,3}\right|^2 + \left|h_{2,1}^* h_{2,6}\right|^2 + \left|h_{4,1}^* h_{4,4}\right|^2 + \left|h_{4,1}^* h_{4,5}\right|^2 \right)
\tag{16}
$$

$$
\sigma^2 = w_{2,1} h_{2,1}^* + w_{4,1} h_{4,1}^*
\tag{17}
$$

Here, $w_{2,1}$ and $w_{4,1}$ are related to $w_{n,k} \in \mathbf{w}$, which shows IID AWGN of the link between *n*th resource and *k*th user.

With an appropriate pulse-shaping approach, PLC-SCMA combined with OFDM can add to the system's spectrum efficiency and resilience to noise and interference. Moreover, selective fading of PLC will appear as flat fading due to OFDM signaling.

*2.5. FFT-SCMA System Model for PLC*

The block representation of FFT-SCMA is illustrated in Figure 5. The incoming bits $b_1, \ldots, b_k$ from $K$ different users are first encoded by using individual codebooks. These encoded symbols $x_{n,k} \in \mathbf{x}$ are superimposed and then passed through the OFDM modulator to prepare the signal for multicarrier transmission. An Inverse Fast Fourier Transform (IFFT) operation is applied to the encoded signal to attain time-domain samples before transmission and is expressed as follows:

$$
\mathbf{x}_f(i) = \frac{1}{\sqrt{M}} \sum_{m=0}^{M-1} \mathbf{x}_{n,k}(m) e^{j2\pi i m / M}, \quad m = 0, 1, \cdots M - 1
\tag{18}
$$

where $\mathbf{x}_f(i)$ denotes the *i*th time-domain sample of $\mathbf{x}_f$ and $M$ indicates the number of subcarriers. The signal received through the PLC channel is denoted as $\mathbf{y}_f$. It is expressed as

$$
\mathbf{y}_f = \mathbf{H}_s' \mathbf{x}_f + \mathbf{w}
\tag{19}
$$

where $\mathbf{H}_s'$ is referred to (4) and $\mathbf{w}$ denotes the IID AWGN noise of each link between the resource and the user, each link being independent of the others as specified in (12).

The received signal is combined through MRC for each user as described in (13). Therefore, $\hat{\mathbf{y}}_\mathbf{f} = IFFT(\mathbf{y}')$ is defined as the IFFT filtered version of (13). Then, FFT is applied after the removal of CP to obtain the following signal $\mathbf{y}_f'$,

$$
\mathbf{y}_f'(l) = \frac{1}{\sqrt{M}} \sum_{i=0}^{M-1} \hat{\mathbf{y}}_f(i) e^{-j2\pi i l / M}, \quad i = 0, 1, \ldots, M - 1
\tag{20}
$$

The received SINR of the FFT-SCMA system is deduced as follows:

$$\text{SINR}_f = \frac{S}{I+N} = \frac{E_s * (h_f')^2}{E_s \, H_f + (h_f')^2 \sigma_f^2} \tag{21}$$

where $h_f' = \frac{1}{\sqrt{M}} \sum_{i=0}^{M-1} h'(i) e^{-j2\pi(m-i)l/M}$ denotes the desired signal, $H_f = \frac{1}{\sqrt{M}} \sum_{i=0}^{M-1} H(i) e^{-j2\pi(m-i)l/M}$ shows IUI, and $\sigma_f^2 = \frac{1}{\sqrt{M}} \sum_{i=0}^{M-1} \sigma^2(i) e^{-j2\pi(m-i)l/M}$ indicates the link noise variance. Furthermore, $H$, $h'$, and $\sigma^2$ are defined in (15), (16), and (17), respectively, for a simple scenario of six users.

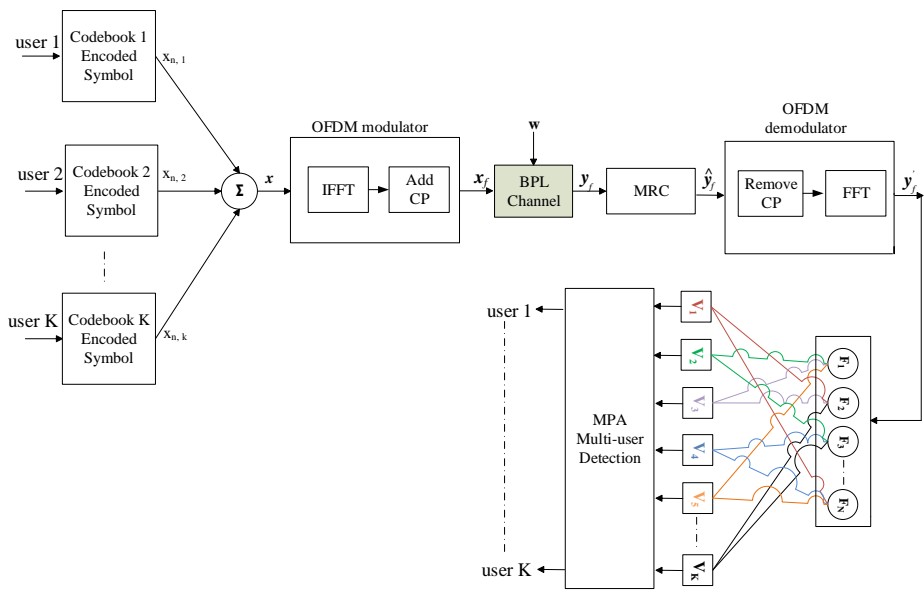

**Figure 5.** Block diagram of FFT-SCMA.

### 2.6. DWT-SCMA System Model for PLC

FFT is suitable specifically for signals whose frequency content does not change with time. DWT has the ability to zoom the time-domain signal in both the frequency and time windows. As a result, it provides better visualization of a signal, which helps in analyzing and reducing the noise content in a signal. The high-frequency components are removed during multiresolution analysis, resulting in less interference and noise in the incoming signal. It improves the received SINR of the proposed system. Moreover, the pulse shape of DWT offers spectral efficiency by eliminating the need for CP and minimum data loss in signal reconstruction as compared with FFT. DWT can characterize a signal accurately. It is utilized efficiently in digital communications for channel modeling, with OFDM modulation, and interference mitigation.

The following equation provides the Continuous Wavelet Transform (CWT)

$$\mathbf{x}_{wt}(\tau, s) = \frac{1}{\sqrt{s}} \int \mathbf{x}(t) \cdot \psi\left(\frac{t-\tau}{s}\right) dt \tag{22}$$

where $\tau$ and $s$ are the translation and scaling parameters, respectively. $\mathbf{x}(t)$ is the signal to be examined, $\psi$ is the mother wavelet or the basis function such as Haar, Daubechies, Symlets, and Coiflets.

The DWT, on the other hand, is simply a sampled version of the CWT. In CWT, the signals are analyzed using a set of basis functions that are related to one another via simple scaling and translation. In the instance of DWT, digital filtering techniques are used to obtain a time-domain representation of the digital signal. The DWT of the discrete type signal $\mathbf{x}[r]$ of length $R$ can be computed using a recursive cascade structure consisting of down-samplers $\downarrow 2$, low-pass ($G$) filters, and high-pass ($H$) filters that are uniquely

associated with a wavelet. The signal is decomposed iteratively through a filter bank to attain its DWT. This results in a novel interpretation of the wavelet decomposition by splitting the signal into frequency bands. At each level, the signal is decomposed into approximated $cA_j$ and detailed $cD_j$ versions of the wavelet function $\Psi(t)$ through low-pass and high-pass filters, respectively. The output of the low-pass filter is used as the input to a new pair of filters in hierarchical decomposition. DWT is applied in the proposed system, which is depicted in Figure 6 (factor graph and decoder is the same as shown in Figure 5). The coefficients of the filters that correspond to scaling and wavelet functions are related by $g[r] = (-1)^r h[L - r], \quad r = 0, 1, \ldots, L - 1$, where $L$ symbolizes the filter length. The filter length and values of approximated and detailed coefficients change amongst wavelet families. The $r$th sample of the SCMA-encoded signal is decomposed as follows:

$$\mathbf{x}_g^j[q] = \sum_{r=0}^{\infty} x[r] g[2q - r] \tag{23}$$

$$\mathbf{x}_h^j[q] = \sum_{r=0}^{\infty} x[r] h[2q - r] \tag{24}$$

The signal before transmission is a combination of each $j$th branch's low-pass filter output and a summed-up version of all $j$ branches' high-pass filters, as shown below.

$$\mathbf{x}_w = \mathbf{x}_g^J + \sum_{j=0}^{J} \mathbf{x}_h^j \tag{25}$$

where $\mathbf{x}_w$ is obtained after taking the Inverse Discrete Wavelet Transform (IDWT) of the encoded SCMA signal and then it passes through the PLC channel, the received signal can be written as follows:

$$\mathbf{y}_w = \mathbf{H}_s' \sum_{m=0}^{M-1} \mathbf{x}_w^m + \mathbf{w} \tag{26}$$

where $M$ denotes the number of the subcarriers and $\mathbf{H}_s'$ is referred to (4). $\mathbf{w}$ is specified in (12) as IID AWGN. The received signal is combined through MRC, which is described as $\hat{\mathbf{y}}_\mathbf{w} = IDWT(\mathbf{y}')$, where $\mathbf{y}'$ is mentioned in (13). The signal is then analyzed through DWT and written below after passing through multilevel filters,

$$\mathbf{y}_w' = \hat{\mathbf{y}}_{wg}^J + \sum_{j=0}^{J} \hat{\mathbf{y}}_{wh}^j + \mathbf{w}_g^J + \sum_{j=0}^{J} \mathbf{w}_h^j \tag{27}$$

$\sigma_w^2 = \mathbf{w}_g^J + \sum_{j=0}^{J} \mathbf{w}_h^j$ is the variance of IID AWGN for the proposed scheme. The received SINR of $k$th user is expressed as

$$\text{SINR}_w = \frac{S}{I + N} = \frac{E_s * h_w^2}{E_s \, H_w + h_w^2 \sigma_w^2} \tag{28}$$

where $H_w = DWT(H) = H_{wg}^J + \sum_{j=0}^{J} H_{wh}^j$ shows interference and $h_w = DWT(h') = h_{wg}^J + \sum_{j=0}^{J} h_{wh}^j$ denotes the desired signal gain. Moreover, $H$ and $h'$ are provided in (15) and (16) for a particular example, respectively.

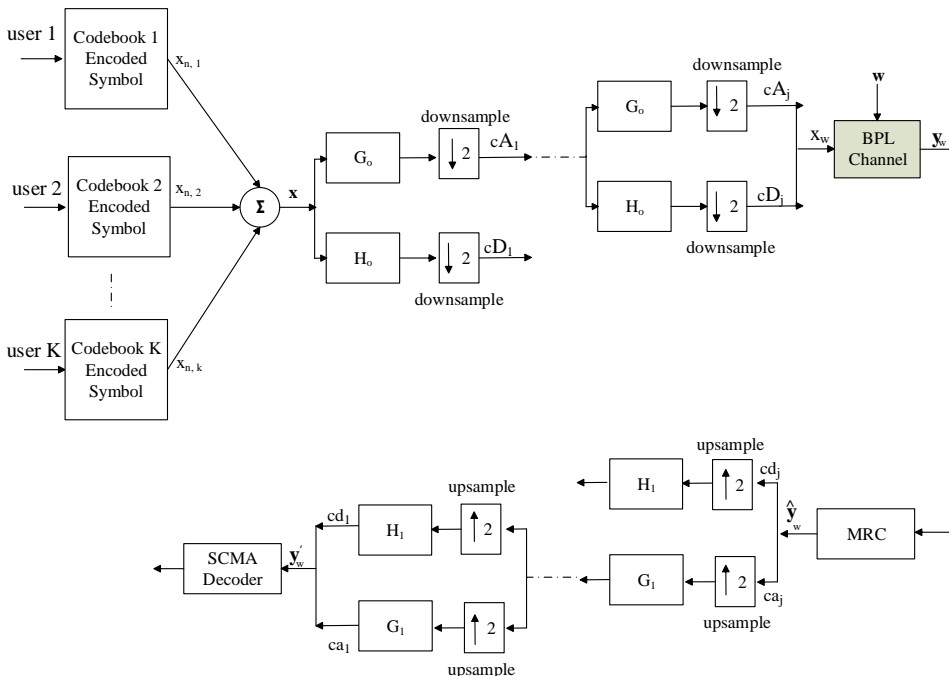

**Figure 6.** Block diagram of DWT-SCMA.

### 2.7. Capacity

The maximum number of bits that may be sent without errors is known as the capacity. Therefore, capacity for PLC-SCMA is calculated as the maximization of mutual information over the Probability-Density-Function (PDF) $p_{\mathbf{x}}$ of all possible transmitted vectors,

$$
\begin{aligned}
C &= \mathrm{E}\left\{ \max_{p_{\mathbf{x}}}\{ I(\mathbf{x}; \mathbf{y} \mid \mathbf{H}_s)\} \right\} \\
&= \mathrm{E}\left\{ \max_{p_{\mathbf{x}}}\{ \mathcal{H}(\mathbf{y} \mid \mathbf{H}_s)\} - \mathcal{H}(\mathbf{y} \mid \mathbf{x}, \mathbf{H}_s) \right\}
\end{aligned}
\tag{29}
$$

where mutual information is further defined in terms of differential entropy $\mathcal{H}$. It is assumed that CSI is known at the receiver; therefore, $\mathcal{H}(\mathbf{y} \mid \mathbf{x}) = \mathcal{H}(\mathbf{H}'_s\mathbf{x} + \mathbf{w} \mid \mathbf{x}, \mathbf{H}'_s)$ will reduce to $\mathcal{H}(\mathbf{w} \mid \mathbf{H}'_s)$ due to $\mathcal{H}(\mathbf{H}'_s\mathbf{x} \mid \mathbf{x}, \mathbf{H}'_s) = 0$. Furthermore, the zero-mean complex Gaussian distribution $\mathcal{CN}(0, \sigma^2)$ is the one that maximizes the entropy. The following form for (29) is obtained by solving the respective differential entropy,

$$
C = \log_2(\pi e \sigma_y^2) - \log_2(\pi e \sigma^2)
\tag{30}
$$

where the variance of $\mathbf{y}$ is provided as $\sigma_y^2 = |\mathbf{H}'_s|^2 P + \sigma^2$, $\sigma^2$ and $P$ refers to variance of IID AWGN and signal power, respectively. After simplification, the final form of capacity for $k$th user is expressed as follows:

$$
C_k = \frac{1}{M + C_p} \sum_{m=0}^{M-1} \log_2\left(1 + \mathrm{SINR}^k * |\mathbf{H}'_s|^2\right)
\tag{31}
$$

where $M$ represents the number of subcarriers and SINR is derived for FFT-SCMA and DWT-SCMA in Sections 2.5 and 2.6, respectively. Moreover, $C_p$ denotes the length of the CP in the FFT-SCMA system, which is omitted in DWT-SCMA and becomes zero.

*2.8. Theoretical Symbol-Error-Rate*

Theoretical values of SER can be computed by taking the Q-function of SINR expressions in Equations (14), (21), and (28), respectively. The SERs for SCMA, FFT-SCMA, and DWT SCMA are given as $\text{SER} = \frac{1}{2}Q\left(\sqrt{\text{SINR}_l}\right)$, $\text{SER}_f = \frac{1}{2}Q\left(\sqrt{\text{SINR}_f}\right)$ and $\text{SER}_w = \frac{1}{2}Q\left(\sqrt{\text{SINR}_\mathbf{w}}\right)$, respectively.

*2.9. Peak-to-Average-Power Ratio*

This section discusses the PAPR of wavelet filter banks in DWT-SCMA. The multicarrier systems based on FFT-OFDM contend with a high PAPR because of the characteristics of IFFT/FFT operations. This signal is a combination of narrowband signals of various tones. When these $M$ subcarriers are combined constructively, the instantaneous power of these peaks exceeds the average signal power. As a result, the peak power of the FFT-OFDM signal is $M$ times higher than the average power. The high PAPR degrades the performance of power amplifiers, resulting in intermodulation distortion and energy inefficiency [42].

The PAPR of a reconstructed signal using IDWT is defined as follows:

$$\text{PAPR} = \frac{\mathcal{P}_{\text{peak}}}{\mathcal{P}_{\text{avg}}} = \frac{\max_n\left\{|\mathbf{x}_w|^2\right\}}{E\left\{|\mathbf{x}_w|^2\right\}} \tag{32}$$

where $\max_n\{.\}$ represents the highest value of time index $n$ among all the indices and $E\{.\}$ indicates the ensemble average calculated over a single symbol's duration. The Cumulative-Distribution-Function (CDF) is given as

$$F_{\mathbf{x}_{\max}}(\mathbf{x}_w) = P(\mathbf{x}_{\max} < \mathbf{x}_w) \quad = (F_{\mathbf{x}_{\max}})^M = \left(1 - e^{\frac{-\mathbf{x}_{\max}}{2E\left\{|\mathbf{x}_w|^2\right\}}}\right)^M \tag{33}$$

As the PAPR of an MC system is determined by the pulse shape, it can be lowered by selecting a suitable pulse shape as $\text{PAPR} \leq M_{\max}|\phi(m)|^2$. Here, $\phi(m)$ symbolizes the scaling function. To characterize the statistical properties of PAPR, a Complementary CDF (CCDF) is commonly defined as $CCDF = 1 - CDF$. For an IDWT frame, CCDF is provided as follows for a constant value $\lambda$,

$$CCDF(PAPR(\mathbf{x})) = 1 - \prod_{m=0}^{M-1} \Pr\left\{|\mathbf{x}_w|^2 \leq E\left[|\mathbf{x}_w|^2\, \text{PAPR}(\lambda)\right]\right\}$$
$$= 1 - \left(F_x\left(\sqrt{E\left[|\mathbf{x}_w|^2\, \text{PAPR}(\lambda)\right]}\right)\right)^M \tag{34}$$

**3. Result Discussion**

To create simulation results in this section, multiplexed users are considered to have equal transmit power. 1TX/1RX antenna configuration is considered for each user with perfect synchronization and perfect CSI. Moreover, the signal is modulated using 4-QPSK, codebooks, and Haar wavelet in the case of DWT-SCMA.

*3.1. SINR Analysis*

SINR improves with interference and noise reduction and is illustrated in Figure 7. As PLC-SCMA supports overloaded transmission, it increases bandwidth efficiency as well as IUI. PLC-SCMA is coupled with OFDM and various pulse-shaping algorithms such as FFT and DWT to increase data transmission reliability for each user. SINR for both cases lies between the worst- and the best-case scenario of PLC-SCMA. The worst case represents the overloaded PLC-SCMA, i.e., with maximum bandwidth efficiency experiencing the highest level of interference due to simultaneous users within the system, as discussed

in Section 2.4. The best, or ideal case, characterizes the least bandwidth-efficient scenario with no interferer on any available frequency resource, i.e., when an intended user utilizes the whole spectrum of a resource with no sharing partner. Figure 7 plots SINR for these techniques, with symbol energy on the x-axis. All cases except the ideal case represent an overloaded scenario with $K > N$. At $1 \times 10^{-13}$ Joule symbol energy, 4.6 dB SINR is obtained for PLC-SCMA, 6.35 dB for FFT-SCMA, 8.13 dB for DWT-SCMA, and 8.80 dB SINR for least spectral-efficient case without IUI. This shows an $\approx$28% rise in the SINR of FFT-SCMA as compared with PLC-SCMA and a 21% increase in SINR values of DWT-SCMA as equated to FFT-SCMA. This trend rises with higher values of $E_s$—for instance, at $2 \times 10^{-13}$ Joule, observed SINR is 6.0 dB, 9.3 dB, and 13.7 dB for three overloaded systems, e.g., PLC-SCMA, FFT-SCMA, DWT-SCMA, respectively, and 16.2 dB SINR for the ideal case with no interferer. From these values, a 35% improvement in SINR for FFT-SCMA as compared with PLC-SCMA and a 32% boost for DWT-SCMA as equated to FFT-SCMA is estimated for overloaded cases while considering signal communication over power lines.

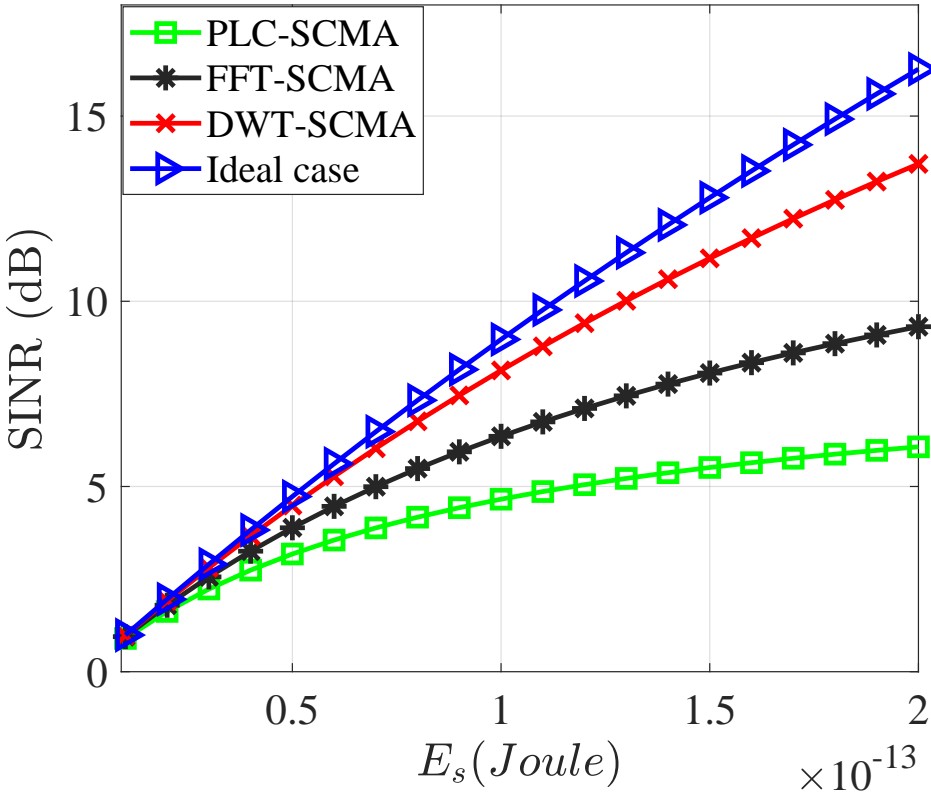

**Figure 7.** SIN vs. Symbol Energy, a comparative analysis among SCMA and FFT/DWT-SCMA.

The PLC-SCMA system experiences the highest IUI $|H|$, which is calculated by using the expression in (16). This IUI is lowered in FFT-SCMA. It gives lower IUI as compared with $|H|$, which is mentioned as $|H_f|$ in Section 2.5. DWT-SCMA achieves higher SINR than FFT-SCMA because the IUI $|H_w|$ decreases as described in Section 2.6. In comparison with FFT-SCMA, DWT-SCMA experiences less interference in the PLC channel due to its particular basis functions for pulse shaping. Interference is low in DWT-SCMA due to the decreased ICI attained with the low-power side-lobes of the subcarriers. PLC research conducted in [43] provides in-home power line channel parameters that are used to calculate channel coefficients for discussed models, and IUI values are measured as illustrated in Table 1.

**Table 1.** Interuser Interference comparison.

| Multiaccess Technique | Interuser Interference |
| --- | --- |
| PLC-SCMA | $|H| = 0.2263$ dB |
| FFT-SCMA | $|H_f| = 0.0230$ dB |
| DWT-SCMA | $|H_w| = 0.0093$ dB |

The wavelet filters produce low out-of-band (OOB) radiation, whereas FFT creates time-limited rectangular-shaped pulses, leading to more spectrum leakage, as shown in Figure 8. Additionally, the suggested DWT-SCMA is more energy-efficient than FFT-SCMA due to the sparse structure of DWT.

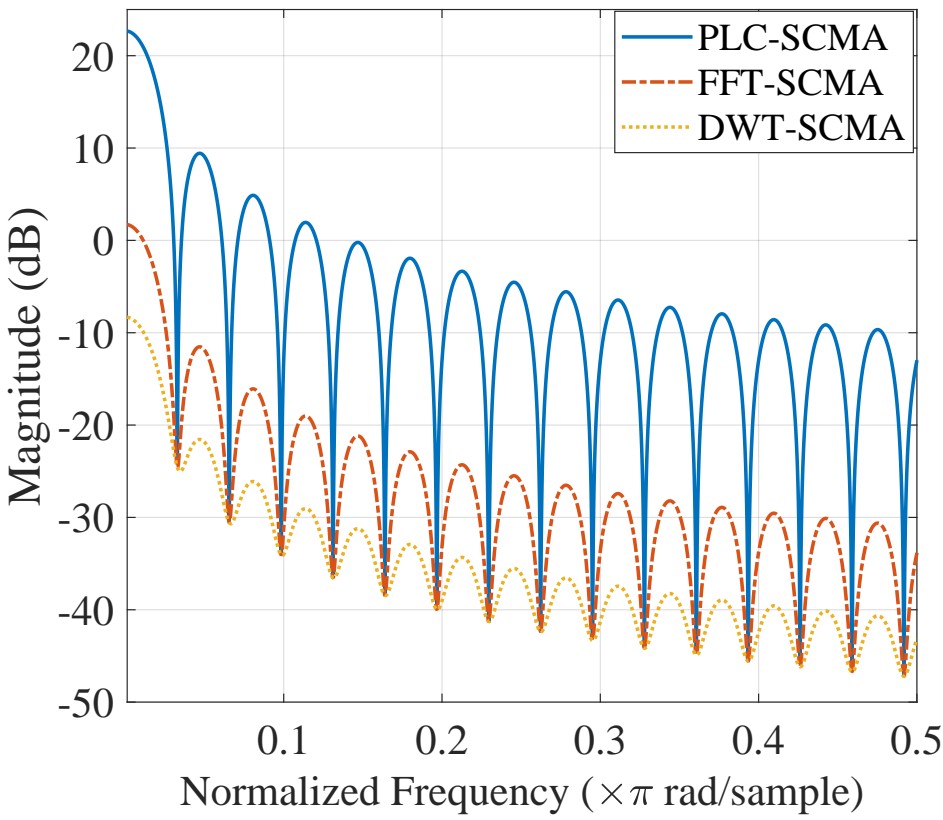

**Figure 8.** Out-of-band radiation comparison among PLC-SCMA, FFT-SCMA, and DWT-SCMA.

### 3.2. Capacity Analysis

Figure 9 depicts the average user capacity analysis of PLC-SCMA, FFT-SCMA, and DWT-SCMA in an overloaded scenario, while the ideal case is drawn when there is no interference with an intended user. The modulated data are transmitted via subcarriers, which reduces the impacts of multipath fading in the PLC channel and increases SINR. Therefore, FFT-SCMA performs better in terms of capacity than PLC-SCMA. To prevent ISI, FFT-SCMA requires 25% bandwidth for CP insertion, which results in bandwidth waste. DWT-SCMA has a lower ICI and does not require CP. It has a higher capacity than FFT-SCMA. This is because more data can be transmitted over the saved bandwidth.

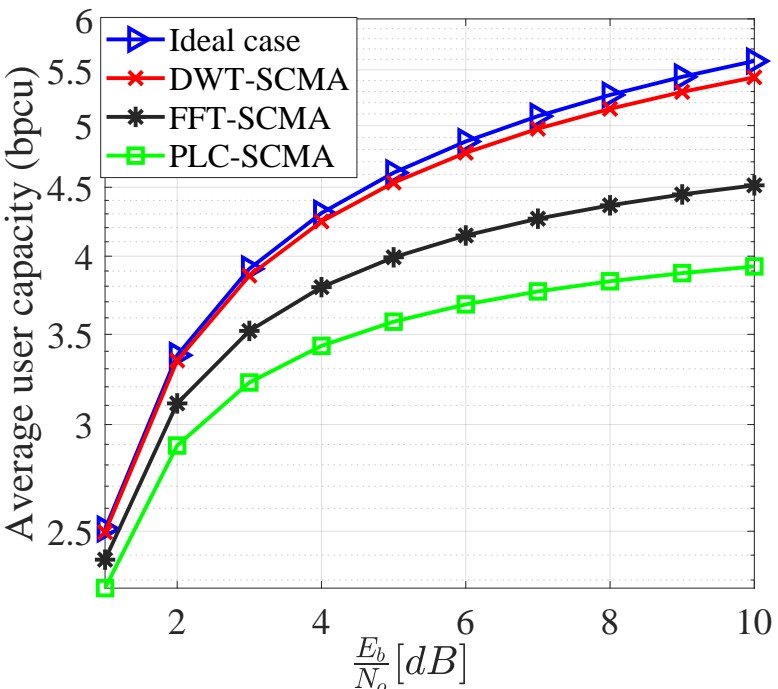

**Figure 9.** Average user capacity vs. $E_b/N_o$, a comparative analysis among PLC-SCMA and FFT/DWT-SCMA.

### 3.3. PAPR Improvement

Figure 10 shows a PAPR performance comparison of FFT-SCMA and DWT-SCMA. As a result of the pulse shape design, it is extrapolated that the probability of a higher PAPR for the same threshold level is higher in FFT-SCMA than in DWT-SCMA. Furthermore, because numerous wavelet families are available, DWT-SCMA provides flexibility in the selection of basis functions based on the transmission characteristics, such as the PLC channel in this study. The higher PAPR of FFT filter-banks-based MC systems compared with wavelet-based communication systems has been demonstrated in several publications [32].

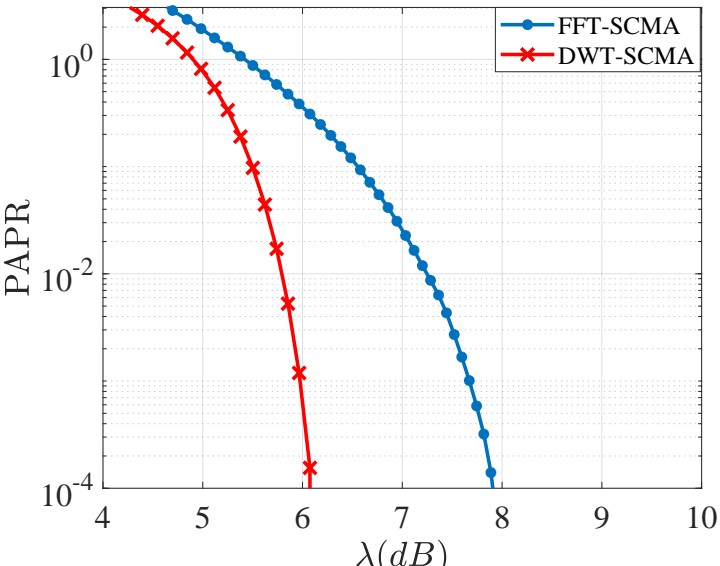

**Figure 10.** Comparison of PAPR performance of FFT-SCMA with DWT-SCMA.

### 3.4. Comparative Analysis of SER

The SER plot with respect to $E_s/N_o$ values for three overloaded cases, e.g., PLC-SCMA, FFT-SCMA, and DWT-SCMA, is illustrated in Figure 11, where $K > N$. The ideal case is drawn as a benchmark performance that depicts a scenario with $K = N$. PLC-SCMA is observed as the worst case due to the noisy PLC channel and high IUI on the frequency resources. An attempt to control the performance degradation parameters using pulse-shaping techniques with flat fading channel effects is made, which protect individual users' data. FFT-SCMA achieves 4-dB gain in SER as compared with PLC-SCMA. It performs better than PLC-SCMA, but due to excessive side-lobe energy in the subchannels, which causes ICI, it falls short of the optimum scenario. DWT-SCMA outperforms FFT-SCMA due to DWT's stronger reconstruction properties and less ICI due to less power in the subchannel side-lobes. Furthermore, DWT-SCMA aids in the reduction of channel noise, and its performance closely resembles that of the ideal scenario. In comparison with FFT-SCMA, DWT-SCMA produces a 6-dB gain in SER.

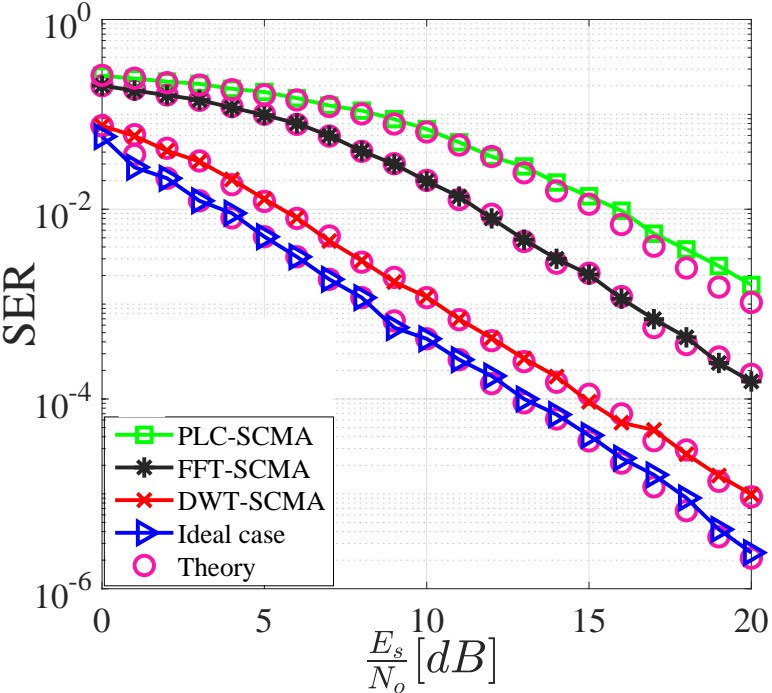

**Figure 11.** SER vs. $E_s/N_o$, a comparative analysis among PLC-SCMA and FFT/DWT-SCMA.

### 3.5. Computational Complexity

Computational complexity is a key consideration while designing any system. The comparison in terms of complex operations is shown in Figure 12. The algorithm that calculates discrete Fourier transform determines the complexity of FFT-SCMA. FFT employs divide and conquer approach and its complexity is $\sim O(N_s \log N_s)$, where $N_s$ is the length of modulated data or number of subcarriers [23]. In DWT-SCMA, on the contrary, wavelet filter banks are used to perform multiresolution analysis. The computational cost of DWT is $\sim O(N)$ if the prototype filter length is negligible compared to the number of signal samples $N$ [23]. For simplicity, the signal samples $N$ are assumed to be equal to the number of subcarriers $N_s$. In comparison with FFT-SCMA, DWT-SCMA has the added benefit of being less complex.

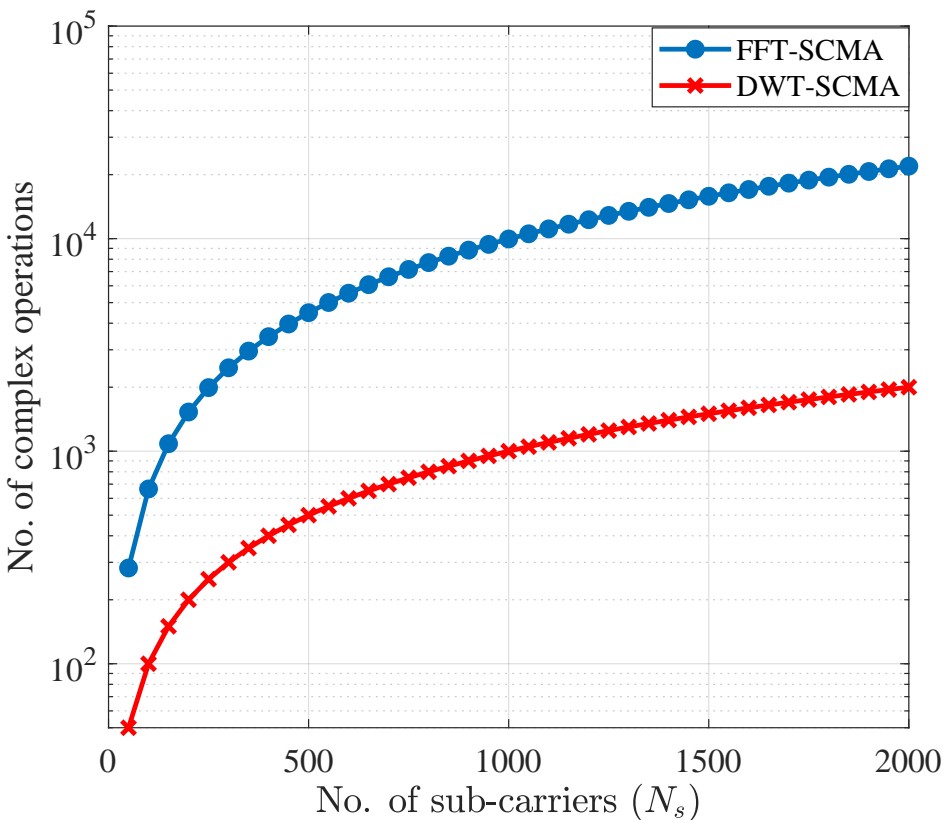

**Figure 12.** Number of complex operations in FFT-SCMA and DWT-SCMA.

## 4. Conclusions

Pulse-shaping techniques such as FFT and DWT were employed in this research to mitigate interference in a PLC-SCMA system. Mathematical and simulation findings demonstrate reduction in interference in DWT-SCMA compared with FFT-SCMA and PLC-SCMA. Performance measurements such as SINR, user capacity, PAPR, and SER have demonstrated that DWT-SCMA outperformed FFT-SCMA in the PLC channel. The SINR increases because DWT helps to reduce PLC channel impairments. The flexibility of wavelet pulse-shaping has reduced PAPR. Due to the absence of CP and low OOB emission, DWT-SCMA has relatively lower overhead and greater spectrum efficiency. Although DWT's sparse structure has reduced complexity, disregarding high-frequency components during the decomposition process could lead to information loss. The DWT-SCMA can be examined for the expected huge number of connections in smart grid applications in future.

**Author Contributions:** M.S.S. contributed in methodology, software, validation, formal analysis, investigation, and writing/editing original draft. S.B. contributed in methodology, investigation, supervision, project administration, and reviewing the original draft. H.M.A. contributed in validation, investigation, supervision, project administration, and reviewing the original draft. K.R. contributed in funding acquisition, project administration, and reviewing the original draft. Finally, S.A.-B. contributed in visualization, project administration, and reviewing the original draft. All authors have read and agreed to the published version of the manuscript.

**Funding:** This research received no external funding.

**Conflicts of Interest:** The authors declare no conflict of interest.

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
