# Peer review of "Wavelet-Transform-Based Sparse Code Multiple Access for Power Line Communication"

_electronics, doi:10.3390/electronics11162618_

Round 1

Reviewer 1 Report

This article adopts the intriguing aspects of DWT to address the interference difficulties. The proposed technique is mathematically modeled and compared to Fast Fourier Transformed SCMA (FFT-SCMA). In the PLC environment, DWT-SCMA is found to outperform FFT-SCMA. The topic is interesting and the writing is well. However, the harmonic is one the most important interfere for PLC, which is reported through soc-based droop coefficients stability region analysis of the battery for stand-alone supply systems with constant power loads. This harmonic should be discussed in the introduction. Furthermore, the innovation points should be rewritten through several point. The comparison simulation and experimental should be added in the revised version.

Author Response

The step by step response is attached as Reviewer1.pdf file (Report notes).

Reviewer 2 Report

The paper proposes discrete waveform transform-SCMA for power line communication. I have some minor corrections.

1. Captions for Figures 9 and 10 need correction.

2. Figures and tables should appear after they are mentioned in the text not before. 

3. The term "where" after the equations should not be capitalized.

4. Indentation formatting is not consistent.

5. There are some formatting and grammar errors. Please proofread.

6. Please explain the different between DWT_SCMA in this paper and the one proposed in reference 26.

7. Some of the equation parameters are not properly explained. A table can be added in an appendix.

8. Abbreviations can be explained in a nomenclature.

9. Please add grids to Figure 1.

10. Please add a decoder presentation in addition to the encoder one for Figure 3.

11. Please add computation time analysis on the same platform for section 2.5 in addition to big O notation comparison.

12. Conclusion section needs to be expanded with the contributions and limitations of this work.

13. Please comment on the latency, user capacity, and energy efficiency of the proposed method.

Author Response

The step by step response is attached as Reviewer2.pdf file (Report notes).

Round 2

Reviewer 1 Report

The comments have been solved

Reviewer 2 Report

Authors have rectified my concerns.